# Evaluating the structure of cognitive tasks with transfer learning

**Bruno Aristimunha**[1,2]    **Raphael Y. de Camargo**[2]    **Walter H. Lopez Pinaya**[3]
**Sylvain Chevallier**[1]    **Alexandre Gramfort**[1]    **Cédric Rommel**[1,4]

[1]Université Paris-Saclay, Inria, France
[2]Federal University of ABC, Santo Andre, Brazil
[3]King's College London, London, United Kingdom
[4]Valeo.ai, Paris, France

## Abstract

Electroencephalography (EEG) decoding is a challenging task due to the limited availability of labelled data. While transfer learning is a promising technique to address this challenge, it assumes that transferable data domains and tasks are known, which is not the case in this setting. This study investigates the transferability of deep learning representations between different EEG decoding tasks. We conduct extensive experiments using state-of-the-art decoding models on two recently released EEG datasets, ERP CORE and $M^3CV$, containing over 140 subjects and 11 distinct cognitive tasks. We measure the transferability of learned representations by pre-training deep neural networks on one task and assessing their ability to decode subsequent tasks. Our experiments demonstrate that, even with linear probing transfer, significant improvements in decoding performance can be obtained, with gains of up to 28% compared with the pure supervised approach. Additionally, we discover evidence that certain decoding paradigms elicit specific and narrow brain activities, while others benefit from pre-training on a broad range of representations. By revealing which tasks transfer well and demonstrating the benefits of transfer learning for EEG decoding, our findings have practical implications for mitigating data scarcity in this setting. The transfer maps generated also provide insights into the hierarchical relations between cognitive tasks, hence enhancing our understanding of how these tasks are connected from a neuroscientific standpoint.

## 1 Introduction

While brain encoding consists in predicting brain activations given a certain stimulus, brain decoding tackles the inverse problem: translating recorded neural activity into its originating stimulus or behavior [1]. This stimulus or behavior can be a visual or auditory element presented, the subject internal mental state (e.g. sleep stage), or the cognitive or motor task being performed during the experiment. Brain decoding has several important applications, such as the diagnosis of neurological disorders [2, 3, 4], the detection of seizures [5, 6], automatic processing of polysomnographic recordings [7] and brain-computer interfaces [8, 9, 10], among others. Electroencephalography (EEG) is a common and affordable way to record the neural activity in this context [11]. It has the benefit of being non-invasive, having very high time resolution compared to functional magnetic resonance imaging (fMRI) and not requiring a complex and costly infrastructure such as magnetoencephalography (MEG). In recent years, there has been an increasing interest in using deep learning (DL) models for EEG decoding [12]. As an example, DL has been shown to be the gold standard when it comes to

NeurIPS 2021 AI for Science Workshop.

automatic sleep stage classification [13, 14, 15] and has also demonstrated impressive performances in brain-computer interfaces [16, 17, 18, 19].

Unfortunately, DL models are notorious for being data-hungry to extract generalizable discriminative representations. This characteristic can be a problem when it comes to EEG decoding since the acquisition of labelled EEG data remains a constraint, resulting in datasets of limited size. Indeed, EEG annotation often requires a specialist to run experiments or visually inspect recorded signals for specific patterns. In addition, EEG signals have a very low signal-to-noise ratio, especially when we compare to other fields of application which were DL thrived, such as image, speech, and text. This characteristic makes EEG decoding even more challenging for DL methods in the context of small datasets.

A common technique in DL for dealing with scarce data scenarios is transfer learning (TL), which consists in applying what you have learned in one context to another [20]. In other words, it can be used to leverage a large dataset to improve the performance in a related smaller dataset, making it a promising technique to alleviate the lack of EEG decoding data. However, TL assumes the knowledge of transferable data domains and tasks, which is not fully understood when it comes to brain data. Indeed, even beyond EEG decoding, understanding hierarchical relations between cognitive tasks remains a core question in neuroscience.

Inspired by the Taskonomy study [21] from the computer vision field, this work investigates the relations between cognitive tasks in an EEG decoding setting. Specifically, we measure the transferability of representations learned by DL models between cognitive tasks. This transferability is measured by pre-training DL models to decode the EEG signals of subjects carrying certain tasks and assessing how well a classifier can reuse the learned representations to decode a subsequent task. We carry extensive experiments with three state-of-the-art decoding models [18, 16, 17] trained and evaluated on two recently released EEG datasets, ERP CORE [22] and $M^3CV$ [23], containing in total over 140 subjects with 11 distinct decoding modalities. This enables us to create transfer maps capturing the relationships between pairs of cognitive tasks, as presented in Figure 1. From an EEG processing perspective, our maps can be used to leverage related datasets for alleviating EEG data scarcity with transfer learning. We show that even with a linear probing transfer method, we are able to boost by up to $28\%$ the performance of some tasks. From a neuroscientific standpoint, our results broaden our understanding of the connections between cognitive tasks. We discover evidence that some decoding paradigms elicit very specific and narrow brain activities, since no other paradigm transfer well into them. On the other hand, the decoding of some cognitive tasks benefits from pre-training on all other paradigms, thus demonstrating that they rely on a broad range of representations.

## 2 Related Works

**Transfer Learning and Taskonomy.** Transfer learning refers to the technique of leveraging knowledge acquired from one source domain and task to enhance the performance in another target domain and task [24, 25]. Domain adaptation, domain generalization and self-supervised learning are hence considered sub-fields of TL under this broad definition [25, 26, 27]. The most common TL approach consists in fine-tuning on a target dataset a model that was pre-trained on a source dataset. In this context, the main factor determining the success or failure of TL is the relationship between source and target domains and tasks [28, 29]. Numerous studies have examined the relationship between tasks for TL purposes [21, 30, 31, 32, 33, 34]. One of the most influential works in this area is Taskonomy [21], which investigates these relationships in the context of computer vision tasks. Although we take great inspiration from it, Taskonomy tries to uncover the relation between *visual learning tasks*, while we focus in relating *cognitive tasks*. In this sense, while Taskonomy works with the same input images, analyzing transfer to different output distributions $P_S(Y) \to P_T(Y)$, our setting is more challenging as we need to transfer between joint distributions of input EEG signals and output decoded stimuli $P_S(X, Y) \to P_T(X, Y)$ (*c.f.* subsection 3.2).

**Uncovering cognitive tasks relations with EEG.** While we study the transfer between different tasks, most EEG decoding works in transfer learning focus on transfering between different subjects performing the same fixed task [35, 36, 37, 38, 39, 40, 41, 42, 43, 20, 44, 45, 46]. They hence enter in the more specific sub-category of domain adaptation and generalization. Beyond the transfer learning framework, some works have also studied the structure of cognitive tasks from the perspective of self-supervised representations [47, 48, 49, 50] and invariances shared by different datasets [51].

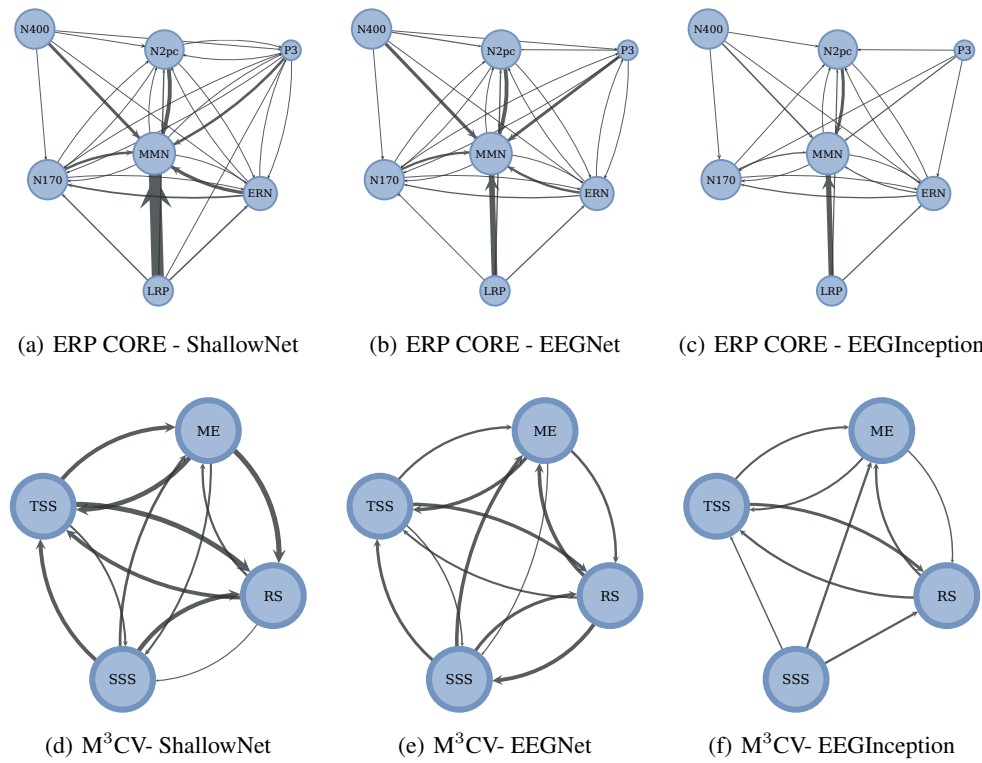

(a) ERP CORE - ShallowNet     (b) ERP CORE - EEGNet     (c) ERP CORE - EEGInception

(d) M$^3$CV- ShallowNet     (e) M$^3$CV- EEGNet     (f) M$^3$CV- EEGInception

Figure 1: Learned tranferability maps for both datasets with three different DL models. These graphs depict the transferability of representations used for EEG decoding, capturing the intricate interplay between cognitive tasks. Each node corresponds to a distinct paradigm. Arrow width represents the average transfer performance when using the representations learned from a source task to decode a target task. Top figures correspond to ERP CORE (1(a), 1(b), 1(c)) and bottom ones to M$^3$CV(1(d), 1(e), 1(f)).

Most related to our work, [52, 53, 34, 54] measure the transferability between cognitive tasks, drawing inspiration from the Taskonomy framework [21]. However, these works are all based on fMRI data and most use *encoding* models for very specific types of visual or language stimuli. In contrast, our work uses EEG *decoding* models to compare a broader range of stimuli in different modalities. By working with decoding instead of encoding models, our results are not only useful to understand the relation between cognitive tasks, but also to improve the performance of automatic EEG processing systems in real-world scenarios where data is scarce.

## 3   Method

### 3.1   EEG Decoding

From a machine learning perspective, EEG decoding is defined as a classification problem, where the outputs $y \in \mathcal{Y}$ correspond to the stimulus or behavior class and the inputs are EEG recordings $\mathbf{x} \in \mathcal{X} = \mathbb{R}^{m \times c}$ with $m$ time-steps and $c$ electrodes. Hence, given a model $f_\theta : \mathcal{X} \rightarrow \mathcal{Y}$ with parameters $\theta$ and a training dataset $\mathcal{E}_{\text{train}} = \{(\mathbf{x}_i, y_i)\}_{i=1}^{N_{\text{train}}}$, training amounts to solving the empirical risk minimization problem:

$$\min_\theta \frac{1}{N_{\text{train}}} \sum_{\mathcal{E}_{\text{train}}} \ell(f_\theta(\mathbf{x}_i), y_i) \ , \tag{1}$$

where $\ell$ is the balanced cross-entropy loss. The performance of the decoding model $f_\theta$ obtained can then be evaluated over a held-out test set $\mathcal{E}_{\text{test}}$ (*c.f.* Figure 2 A, B, C).

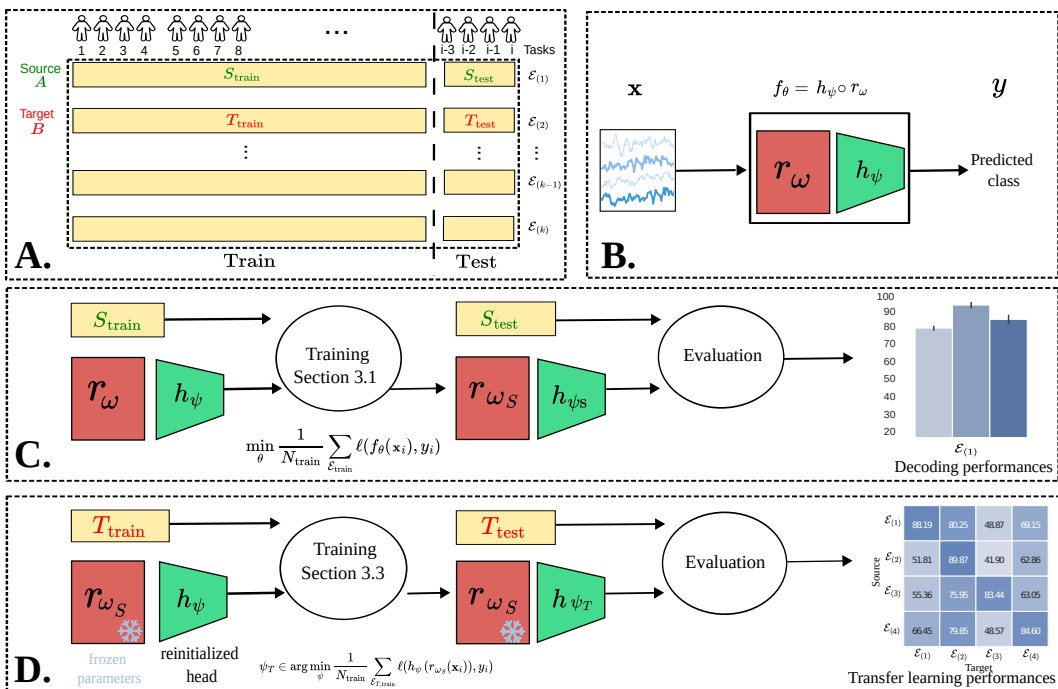

Figure 2: **A.** Data splitting and alignment. Source and target tasks correspond to different ERP and BCI paradigms; **B.** EEG decoding models as a representer network and a classification head; **C.** Standard EEG decoding training and evaluation; **D.** Transfer with linear probing. Only the classification head $h_\psi$ is re-trained, while the representer network $r_{w_S}$ trained on the source task is kept intact.

## 3.2 Transfer learning

More generally, we assume that our dataset corresponds to observations of a joint random variable $(X, Y)$ valued on a space $\mathcal{X} \times \mathcal{Y}$, where some decision function $f$ exists such that $Y = f(X)$. Following [25], a *domain* is defined by a feature space and inputs distribution $\mathcal{D} = \{\mathcal{X}, P(X)\}$. Likewise, a task consists of an output space and a decision function $\mathcal{T} = \{\mathcal{Y}, f\}$. Transfer learning aims to use the knowledge extracted from a source domain $\mathcal{D}_S$ and task $\mathcal{T}_S$ to improve the performance of a model $f_\theta$ on another target domain $\mathcal{D}_T$ and task $\mathcal{T}_T$. In practice, domains and tasks are materialized by source and target datasets $\mathcal{E}_S = \{(x_i, y_i) \sim P_S(X, Y); Y = f_S(X)\}$ and $\mathcal{E}_T = \{(x_i, y_i) \sim P_T(X, Y); Y = f_T(X)\}$.

In our context of EEG decoding, the source and target datasets correspond to different cognitive tasks performed by the same cohort of subjects in comparable experimental settings (*c.f.* section 4). Hence, while source and target domains share the same feature space $\mathcal{X} = \mathbb{R}^{m \times c}$, they differ in terms of marginal distributions of input trial recordings $P_S(X) \neq P_T(X)$. Regarding learning tasks, source and target datasets differ in terms of decision functions $f_S \neq f_T$ and output spaces $\mathcal{Y}_S \neq \mathcal{Y}_T$. As opposed to our setting, Taskonomy [21] works with a common source and target domains $(\mathcal{X}_S, P_S(X)) = (\mathcal{X}_T, P_T(X))$.

## 3.3 Transferability through linear probing

We evaluate the transferability between source and target datasets through *linear probing* [55, 56], as described below. We assume that $f_\theta = h_\psi \circ r_\omega$ is a neural network made of two parts: a representer model $r_\omega : \mathcal{X} \rightarrow \mathcal{R}$, with parameters $\omega$, and a classifier head $h_\psi : \mathcal{R} \rightarrow \mathcal{Y}$, with parameters $\psi$. While the representer $r_\omega$ is responsible for learning useful representations for the learning task, the classifier head $h_\omega$ is just the last linear layer of the network used to deliver the classification decision. When assessing the transferability between a source and target datasets (*c.f.* Figure 2):

1. We first train the whole model $f_{\theta_S} = h_{\psi_S} \circ r_{\omega_S}$ on the training split of the source dataset $\mathcal{E}_{S,\text{train}}$ by solving equation (1) ;

2. Then we freeze the representer parameters $\omega_S$ and retrain the classifier head $h_{\psi_T}$ from scratch on the training split of the target dataset $\mathcal{E}_{T,\text{train}}$:

$$\psi_T \in \arg\min_{\psi} \frac{1}{N_{\text{train}}} \sum_{\mathcal{E}_{T,\text{train}}} \ell(h_{\psi}\left(r_{\omega_S}(\mathbf{x}_i)\right), y_i) ; \qquad (2)$$

3. Finally, we evaluate the network obtained $h_{\psi_T} \circ r_{\omega_S}$ on the test split of the target dataset $\mathcal{E}_{T,\text{test}}$ and use this metric to assess the transferability between $S$ and $T$.

We evaluate the transferability through linear probing since it assesses whether the representations learned from the source dataset allow us to easily (linearly) classify the target data. In contrast, fine-tuning (*i.e.* pushing the training of the whole model $h$ and $r$ further with target data) would modify the learned source representations, which could complicate the analysis [57].

## 4 Experiments

**ERP CORE and M$^3$CV datasets.** During EEG decoding experiments, subjects perform cognitive tasks with stimuli that evoke specific brain signatures. When the nature of the stimuli is similar, they can be categorized into the same paradigm. EEG decoding studies have been interested in a very large and diverse number of paradigms, which can be categorized as exogenous (where an external stimulus is used *e.g.* event related potential) or endogenous (where stimuli are induced by a predetermined mental task or behavior, *e.g.* motor imagery) [58, 8]. Only a small number of EEG datasets contain recordings in a diverse set of paradigms with the same subjects and configurations. Most existing datasets include a limited number of subjects [59, 60] or a limited number of cognitive tasks [61, 62, 63].

In our study, we use two of the few EEG datasets that explore a large diversity of paradigms with a single large cohort of subjects in comparable experimental settings. The first dataset, ERP CORE [22], comprises 40 subjects (25 females and 15 males between 18 and 30 years old). It focuses on exogenous paradigms, featuring seven isolated tasks eliciting specific event-related potential components (ERP), namely Active Visual Oddball (P3b), Word Pair Judgement (N400), Face Perception (N170), Passive Auditory Oddball (MMN), Flankers (LRP and ERN) and Simple Visual Search (N2pc). These tasks are represented in Figure 3 and explained in further details in subsection A.1.

The second dataset, M$^3$CV [23], is a large multi-task, multi-session, multi-subject investigation of EEG commonality and variability. It includes 106 subjects who performed specific tasks in six different paradigms. In this work, we only focus on trials for which the subject and task labels were available, namely: Motor Execution (ME), Transient-State Sensory (TSS), Resting-State (RS) and Steady-State Sensory (SSS). This reduces our dataset to 95 subjects (22 females and 73 males between 19 and 24 years old) corresponding to the enrolment and calibration subsets of the original dataset.

For this dataset, the definition of labels was straightforward. Decoding labels for the *RS* task simply corresponded to whether the subjects had their eyes open or closed. For the *TSS* and *SSS* tasks, labels were directly defined based on the stimulus presented to the subjects, *i.e.* whether it was a visual, auditory or somatosensory stimulation (3-class classification problem). Likewise, decoding labels of the *ME* task corresponded to the movement being executed by the subjects, *i.e.* either the right foot, the right hand or the left hand (3-class classification problem).

**Data splitting.** The datasets were randomly split into a training, a validation, and a test set with respective proportions of 56%, 24% and 20%. Each split contains data from different subjects since we want to assess the cross-subject generalization of our models. Trainings and evaluations were repeated with different splits following a 5-fold cross-validation scheme.

In the standard decoding experiments (subsection B.1), models were trained and evaluated using data from the same cognitive task (*c.f.* Figure 2 C). In the transfer learning experiments (section 5), models were pre-trained on training subjects of some task $A$, then fine-tuned through linear probing

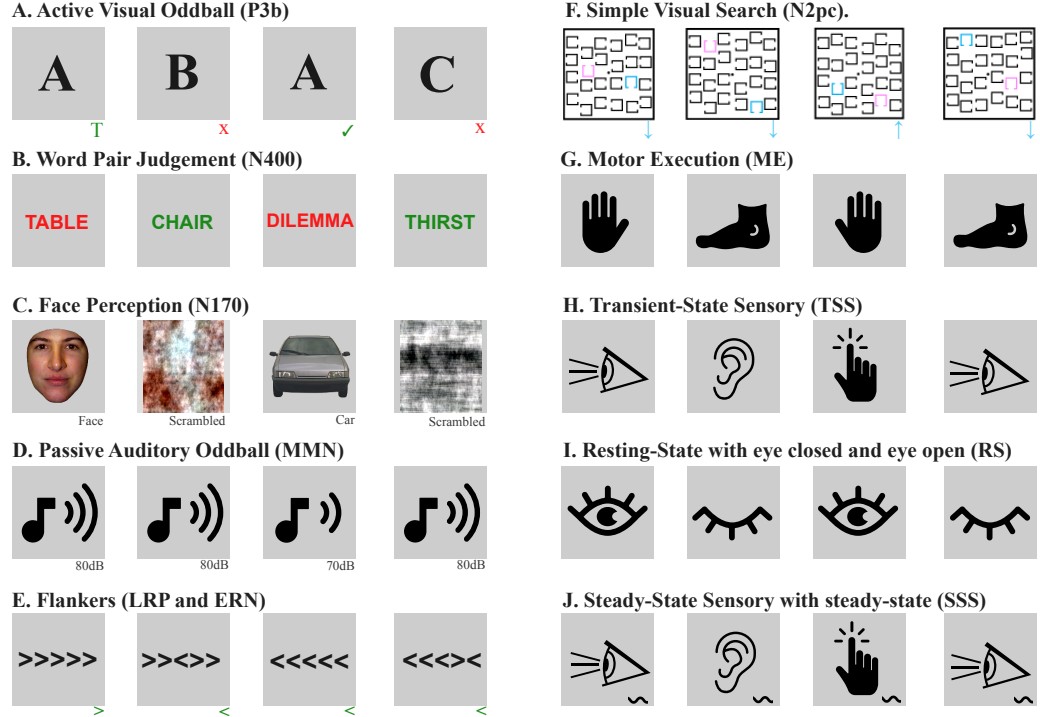

Figure 3: Illustration of experiments recorded in ERP CORE (**A** to **F**) and M$^3$CV datasets (**G** to **J**). A detailed description of these tasks can be found in subsection A.1.

on data from the same subjects performing a different task $B$ and finally evaluated on unseen test subjects carrying out this same task $B$, as described in subsection 3.3 and illustrated in Figure 2 D. Thanks to this data splitting alignment across cognitive tasks, we ensure that the test subjects remain the same after transfer, avoiding any leakage between test and training sets.

**Decoding models.**    We evaluated the transferability of learned tasks using three state-of-the-art deep learning models: ShallowNet [18], EEGNet [16], and EEGInception [17]. These architectures delivered state-of-the-art results in several EEG decoding tasks and datasets [64, 65, 66, 67, 68, 69], and were assessed in a standard setting without transfer in the appendix (subsection B.1). In all our experiments, we used the implementations available in the BRAINDECODE library [18] with default hyperparameter values. More training and preprocessing details can be found in Appendix A.

## 5    Results and Findings

**ERP CORE**    We now analyse the transfer performance between paradigms to quantify how transferable each cognitive task is in relation to one another. Figure 4(a) shows the obtained balanced accuracy for each pair of source and target cognitive tasks in the ERP CORE dataset. A first intriguing observation is the asymmetry of the transfer matrices obtained, regardless of the model used, indicating that the transferability is highly directional. We can see that some tasks do not transfer well, leading to accuracies close to chance ($50\%$ in this dataset). Notably, no source task is correctly transferring either to LRP or N400 paradigms. This means that these cognitive tasks rely on very specific representations that are not elicited by the other tasks and are hence erased by models pre-trained on other paradigms.

Conversely, representations learned on LRP seem useful for decoding ERN and MMN, as they lead to accuracies comparable to a model fully trained and evaluated in a standard fashion (diagonal values). This suggests that LRP elicits brain activations common to ERN and MMN, which are learned by the decoding networks into its hidden representations. More strikingly, pre-training on some source tasks (*e.g.* ERN) improves performance on some other target tasks (*e.g.* N170),

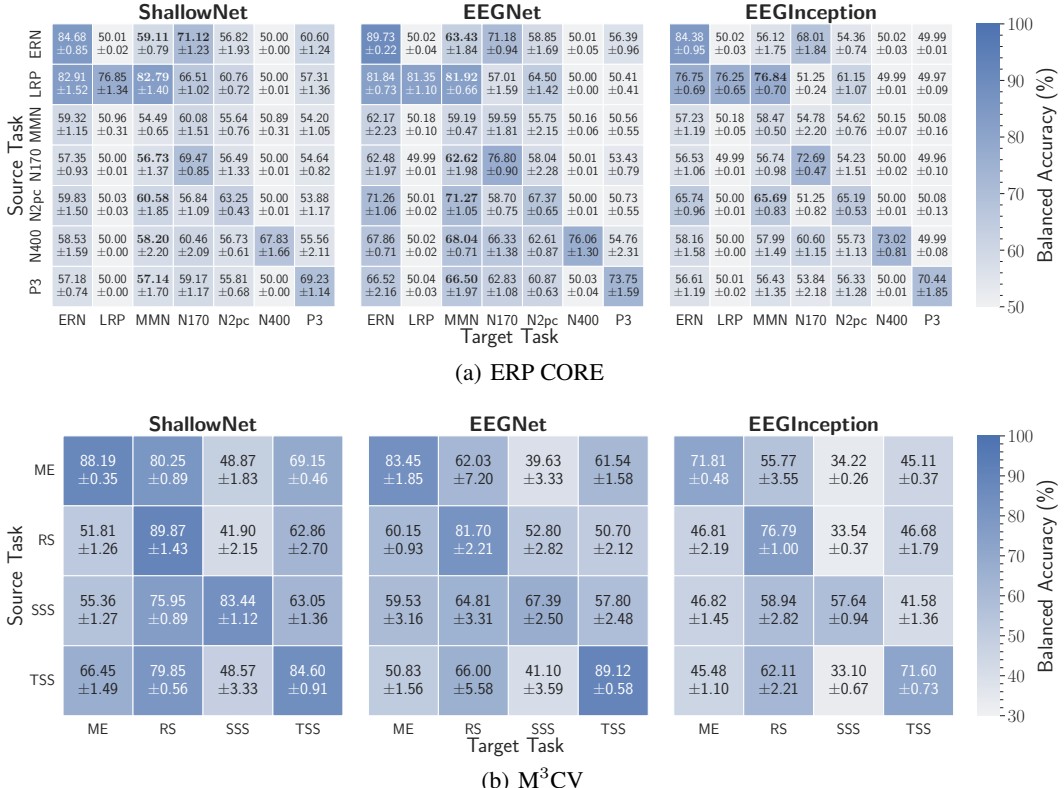

(a) ERP CORE

(b) M$^3$CV

Figure 4: Transfer balanced accuracy for each pair of sources and target cognitive tasks in the ERP CORE and M$^3$CVdataset. Cell values correspond to average performance and standard-deviation across 5-fold cross-validation. Diagonal values correspond to standard decoding balanced accuracies, without any transfer (*c.f.* Figure 2 C).

even *exceeding* the reference performance of a model fully trained on this target task. The best example of this is the MMN task, whose performance is boosted beyond the reference accuracy when transferring *from all possible source tasks*. With ShallowNet we observe improvements of $28.3\%, 6.1\%, 4.6\%, 3.7\%, 2.7\%$, and $2.2\%$ for the LRP, N2pc, ERN, N400, P3, and ERN sources.

In order to better visualize these complex connections between tasks, we processed the matrices from Figure 4(a) into transferability maps, shown in Figure 1. Arrows' widths are proportional to corresponding transfer accuracies, where performances close to chance level were omitted (*c.f.* subsection A.5 for details). The figure clearly shows that LRP is a very good source task and that MMN is a great target task. Note that all three different architectures lead to consistent maps overall, evidencing the stability of our findings.

**M$^3$CV.** Similar asymmetrical transferabilities were obtained for the M$^3$CṼdataset (*c.f.* Figure 4(b)), the best example being SSS and TSS tasks, which share the same class labels: visual, auditory or somatosensory stimuli. Surprisingly, while SSS transfers well to TSS, the opposite is not true, showing that steady-state signals have a specific footprint uncaptured by transient state representations.

From a paradigms perspective, we can see that no transfer accuracies exceed the reference diagonal values in this dataset. Nonetheless, RS appears as a great target task, benefiting from representations learned in all source tasks. To a smaller extent, TSS also exhibits good performances for all source tasks when using the ShallowNet architecture. We hypothesize that this is only visible for this architecture because it is the only one capable of extracting the information shared by all 4 paradigms, given its superior decoding performance for this dataset (*c.f.* subsection B.1). On the contrary, it appears that it is very difficult to transfer onto the SSS paradigm, the extreme case being obtained with EEGInception, for which the chance accuracy is attained for all sources (33% here).

The consistently strong connections towards RS and TSS from all neighboring tasks are more easily seen on the transferability maps of Figure 1, as well as the aforementioned SSS↔TSS asymmetry.

## 6   Discussion and impact

Concerning the potential applications, cognitive maps such as Figure 1 can significantly enhance and optimize the training of predictive models. As exemplified in this study, one compelling approach is to leverage a task that exhibits strong transferability as a template for decoding challenging cognitive tasks, as shown for MMN or N170 ERPs. This aspect proves especially crucial in psychophysical experiments, where subject numbers are limited while experimental variables are abundant. These factors seem to be observed in other areas, such as computer vision [70].

Additionally, the cognitive taxonomy maps can play a pivotal role in the design of improved Human Machine Interfaces (HMIs) for Brain-Computer Interface (BCI) applications. Our focus particularly centers around integrated BCIs [8], which extend beyond mere control functions and delve into scenarios involving error detection and negative potential. By incorporating insights gained from cognitive mapping, we can significantly enhance the performance and effectiveness of HMIs in such contexts. For example, the detection of ERN could help to mitigate situations where the BCI system selects unwanted commands, resulting in user frustration.

Further discussion on the potential impact of this work for clinical applications, source localization, better functional networks understanding and BCI illiteracy mitigation can be found in Appendix C.

## 7   Conclusion

Our study investigates the transferability of deep learning representations through extensive experiments on two EEG datasets. To our knowledge, our work is the first to construct representation transfer maps for EEG decoding. These maps reveal a complex and asymmetric hierarchical relationship between cognitive tasks, enhancing our understanding of brain decoding and neural representations. Our findings also have very practical implications for mitigating data scarcity, demonstrating performance improvements in real world data.

One limitation of our work is that we focus on ERP and BCI datasets. Future work could hence extend our analysis to a broader range of cognitive tasks, or even investigate the transferabily across datasets that do not share the same cohort of subjects. Other interesting avenues for investigation could be the study of other types of transferability, for example by carrying out a similar analysis in a self-supervised setting.

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

# A  Further experimental settings

## A.1  Detailed description of ERP CORE decoding labels

The *Flankers* task aims to elicit two possible ERP components: *ERN* and *LRN*. ERN characterizes situations in which the subject makes an error, even when they are not consciously aware of them [71]. It is elicited here by asking subjects to indicate whether they saw a left or right trigger. We assigned one label to trials where the subject's response matched the target stimulus, and another label when it did not. If the trigger and response were both left or right, the trial was labelled as correct; otherwise, it was incorrect [72, 22, 73].

The same Flankers task is used to elicit the *LRN component*, which is related to the side of the response hand. Unlike ERN, this component is associated with preparing the response rather than its execution, as reported in previous studies [72, 73]. Hence, we relied solely on the subject's response to determine the label for each trial in this case. Specifically, if the subject answered with the left hand, we assigned one label; if they responded with the right hand, we assigned a different label.

The *Word Pair Judgment* task is used to elicit the *N400* component. It consists in presenting participants with a prime word, followed by a target word and asking them to judge whether they are semantically related or not [74]. To define the label for each trial, we use the stimulus definition. Specifically, if the presented words were semantically related, we assigned one label; if they were unrelated, we assigned a different label.

The *N2pc* component is elicited through a *Simple Visual Search* task [75]. In this task, participants had to visually identify a specific target item (in blue or pink) among multiple distractors (in black). Notably, each item is an outlined square with a gap on top or bottom, and subjects had to indicate the position of the gap on the target item (*c.f.* Figure 3). Despite the task assigned to the subjects, what is decoded in this experiment is the position of the target within the screen, *i.e.* whether it appeared on the left or right side, since this is what the N2pc component really captures. As usually done for this experiment [76], we excluded trials with incorrect responses to ensure that the subjects were paying attention to the target.

The *N170 component* [77] was elicited by presenting stimuli in the form of cars, faces, and deformed versions of these objects, in what is called the *Face Perception* task. Participants had to identify whether the presented stimulus was intact or scrambled. To simplify the analysis, we excluded scrambled stimuli as recommended by the original authors of ERP CORE [76], and labelled the remaining ones according to whether they represented a car or a face.

Concerning the *Active Visual Oddball* task, participants were asked to watch a sequence of random letters among A, B, C, D or E. The first letter in a trial had the role of target, and subjects had to answer whether the following letters matched the target or not, which should elicit the *P3b component* [78]. To assign trial labels for this task, we considered two factors. First, we checked whether the letter on the screen matched the target letter. Second, we verified whether the participant provided the correct answer or not. If both conditions were satisfied, we assigned one label to the trial. If not, we assigned another label.

Finally, the *Passive Auditory Oddball* task aims to elicit the *MMN component* [79]. In our study, the stimuli were used to define the trials labels. Specifically, we assigned one label if the individual heard a standard tone at 80 dB and another label if they heard a deviant tone at 70 dB.

## A.2  EEG pre-processing and epoching

Both datasets were pre-processed following the authors' recommendations in all our experiments. Namely, ERP CORE recordings were filtered between $[0.5 - 40]$ Hz with overlap-add FIR filtering. Electric potentials were referenced on the average of electrodes P9 and P10 for all tasks except N170, for which we used the average of all 33 electrodes as commonly done in the literature [22]. Stimuli events were shifted 26 ms forward in time to account for the LCD monitor delay in the MMN task. We downsampled the data from 1000 Hz to 250 Hz and used ICA [80] to correct artefacts and discard particularly bad trials. Finally, trials were cropped into 1000 ms windows based on the stimulus or response depending on the task, following values reported by [22]. This dataset was entirely pre-processed using the MNE-PYTHON library [81].

For the M$^3$CVdataset, we used the dataset pre-processed by the authors as described below. Signals were band-pass filtered between $[0.01 - 200]$ Hz using a 4th order Butterworth filter and notch filtered between $[49 - 51]$ Hz. Potentials were referenced on the average of electrodes TP9 and TP10. Visual inspection and ICA were used to remove artefacts, and bad channels were replaced by the average of the three neighbouring channels. Finally, $1000$ ms signals were cropped from each trial. More pre-processing details can be found in [23].

### A.3 Training setting

All DL models were initialized with the Xavier sampling [82] and trained with the AdamW optimiser [83] using default parameters $\beta_1 = 0.9$ and $\beta_2 = 0.999$ and a weight decay of $5 \times 10^{-4}$. The initial learning rate was set to $10^{-4}$ for ShallowNet and EEGInception, and to $6.25 \times 10^{-4}$ for EEGNet. Trainings lasted at most 200 epochs and an early stopping [84] procedure with a patience of 50 epochs was used. The training of all deep learning models was carried out using PYTORCH and BRAINDECODE libraries [85, 18] on an Nvidia DGX with 4 A100 boards.

### A.4 Baseline models

In our standard decoding experiments presented in subsection B.1, we had to use a selection of baseline models to assess the decoding performance of DL models. As baselines, we tested Riemannian methods, which are known as the best machine learning techniques for BCI and ERP, together with DL models. To this end, we evaluated both Minimum Distance to Riemannian Mean (MDM) [86, 87] and Tangent Space methods with either logistic regression or support vector machine classifiers. We also evaluated these methods with a varied set of covariance matrices, such as ERPCovariances [88] and Xdawn [89]. These methods were implemented and trained using the MOABB and PYRIEMANN library [90, 91].

### A.5 Transferability maps

More precisely, they correspond to transferability scores $s_{S,T}$ for each source and target tasks, obtained by linearly rescaling the corresponding accuracies $a_{S,T}$ so that $0$ corresponds to the chance level $c_T$ and $1$ corresponds to the reference accuracy without transfer (*i.e.* matrix diagonal $a_{T,T}$):

$$s_{S,T} = \frac{\max(a_{S,T} - c_T, 0)}{a_{T,T} - c_T} \, . \tag{3}$$

The Scalable Force-Directed Placement (SFD) layout algorithm [92] was used to create the graphs.

## B  Further experimental results

### B.1 Decoding perfomance

We first analyse the decoding performance of studied models on ERP CORE and M$^3$CVdatasets in a standard decoding setting, without any transfer. Overall, DL methods deliver superior balanced accuracies than baseline methods, as shown in Figure 5.

Their scores also exhibit lower standard deviations than machine learning methods, demonstrating the stability of learned representations across subject splits. Notably, EEGNet consistently led to the best performance on ERP CORE for all paradigms, whereas ShallowNet outperformed others on three out of four paradigms of the M$^3$CVdataset. Also, note that our results on ERP CORE are consistent with those reported in [93, 94].

To support this analysis, we performed a permutation signed-rank test for each model pair within each paradigm to determine whether observed performance gaps are significant (Figure 6). The resulting p-values were combined using Stouffer's method, with a weighting given by the square root of the number of subjects, and a Bonferroni correction was applied to account for multiple comparisons, as done in [90]. The standardized mean difference was calculated within each dataset to determine the effect size. Examining Figure 5(b), we observe that EEGNet slightly outperforms ShallowNet, although the difference is not statistically significant, as highlighted in Figure 6(b).

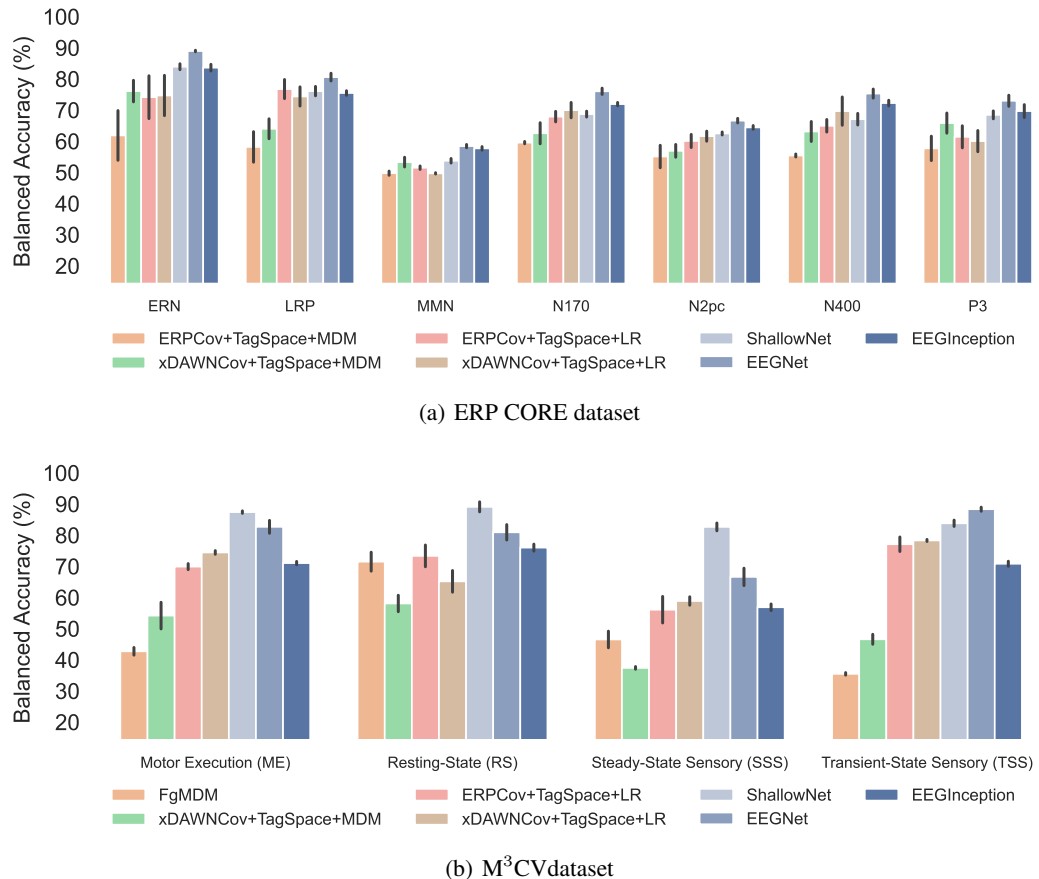

(a) ERP CORE dataset

(b) M³CVdataset

Figure 5: Cross-subject balanced accuracy across paradigms. Error bars correspond to the standard-deviation across 5-fold cross-validation. DL methods outperform machine learning baselines on most paradigms. EEGNet is consistently better than other architectures in ERP CORE . ShallowNet leads to the best scores across M³CVparadigms, except for TSS.

From a paradigms perspective, the most challenging task within the ERP CORE dataset is the MMN paradigm, which is unique in that the subject does not have an active response moment. Conversely, the least difficult tasks correspond to the ERN and LRP.

Overall, the best DL architectures (EEGNet for ERP CORE and ShallowNet for M³CV) lead to similar performances across paradigms, even in scenarios with different numbers of classes (*e.g.* two classes in RS vs. three classes in ME, SSS and TSS). This suggests that the results are robust and confirms that the selected decoding architectures are well-suited to the different experimental paradigms studied in this work.

## B.2    Different transferabilities for EEGInception in M³CV dataset

A striking observation regading Figure 4(b) is that the EEGInception matrix differs from ShallowNet and EEGNet ones, which was not the case when analysing ERP CORE results. We hypothesize that this is probably due to the lower pre-training performance of this model seen on Figure 5(b), evidencing that it learned weaker representations. Interestingly, we see on both ERP CORE and M³CVresults that transfer performances across models (Figures 4(a) and 4(b)) rank consistently with pre-training performances (Figure 5).

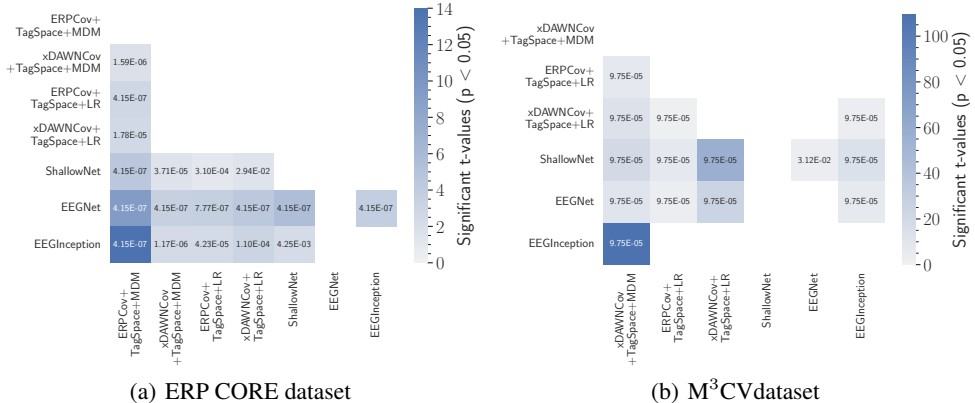

(a) ERP CORE dataset  (b) M³CVdataset

Figure 6: Significant pairwise standardized mean performance difference between decoding models from Figure 5. Statistical significance was computed using a corrected permutation test, with a cutoff p-value of 0.05.

## B.3 Dendograms

As another way to compare tasks, we also clustered them as done in [21] by representing each task by its row vector in the transferability matrix rescaled through (3). This led to the dendrograms depicted in Figure Figure 7. By comparing tasks through their row vectors, we are trying to see whether they transfer in a similar fashion to other tasks.

**ERP CORE**  Most tasks cluster together, with N400-N2pc and N170-ERN being the closest pairs. We also see that LRP stands outside otransferable each cognitive task is in relation to one another. Figure 4 shows the obtained balancedf the cluster, as it is a particularly good source task.

**M³CV.**  The similarity between the best target tasks, RS and TSS, becomes even more apparent on the clustering dendogram from Figure 7(b) Indeed, these paradigms are grouped in their own cluster with EEGNet and appear as the closest paradigms with ShallowNet and EEGInception. This figure also confirms that SSS is a very particular task, as it is outside the main cluster for two out of three dendograms.

## C  Further discussion on impact and applications

The clinical applications of cognitive mapping hold immense potential, especially in aiding patients with schizophrenia who encounter difficulties in emotion detection and facial recognition [96]. Leveraging the insights derived from the N170 task [97], we can devise interventions that boost performance and facilitate improved outcomes for individuals grappling with these challenges. This is especially useful in the context of remediation of emotion recognition, where schizophrenic patients have difficulties to recognise faces and to extract relevant cues (like eyebrows or mouth movements) for interpreting social interaction adequately. Objective information indicating that the patient has recognised a face during an interaction, instead of focusing on irrelevant details for instance, could help therapists for rehabilitation purposes.

Moreover, our findings shed light on the functional networks underlying related tasks. Notably, closely related tasks often engage similar functional networks, suggesting the presence of shared evoked components [72, 73]. This implies that activations observed in one task can enhance and refine the performance of another task. While these tasks may exhibit distinct temporal dynamics and spatial orientations, they ultimately serve the same overarching purpose and engage overlapping cortical networks. Unraveling the sources responsible for generating event-related potentials (ERPs) becomes more feasible when considering these perspective-dependent variations in activation patterns. This understanding illuminates the intricate interplay among different components of the cortical network.

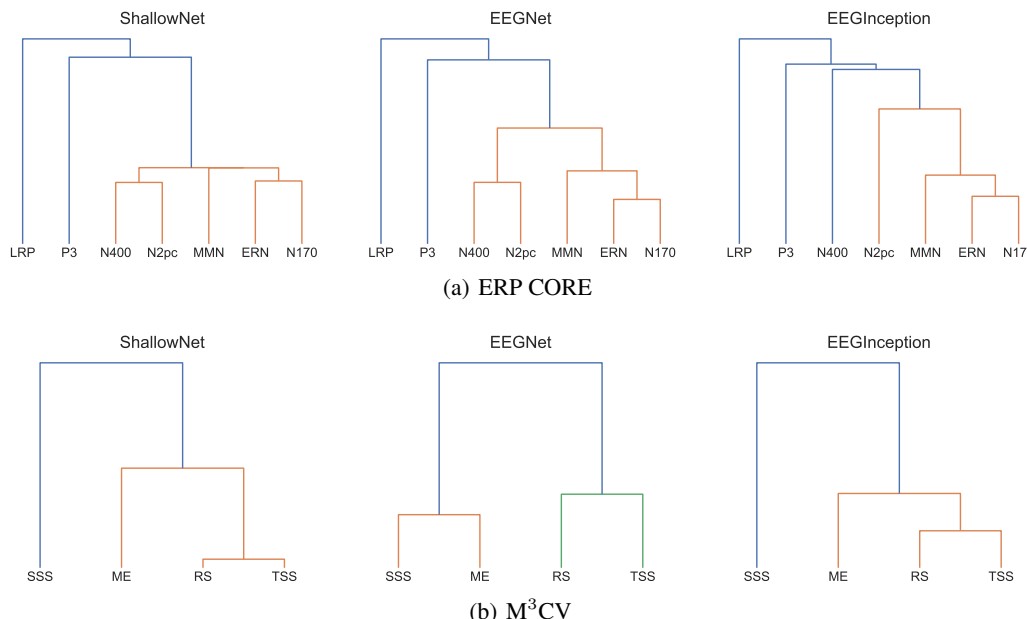

(a) ERP CORE

(b) M$^3$CV

Figure 7: Hierarchical clustering dendrogram, showing the resemblance between source tasks in both datasets. Each task is represented by its row in the transferability matrices (Figures 4(a) and 4(b)) rescaled through Equation 3. Pairwise distances were than computed using the Euclidean distance, before clustering the tasks with the UPGMA algorithm [95].

One compelling avenue for applying the insights gained from our study lies in the realm of source localization, a classic inverse problem in neuroscience [81]. Cognitive mapping can serve as a potent constraint in reconstructing brain activity, leveraging functional regularization. Source localization methods rely on physical and anatomical constraints to yield plausible solutions. Information regarding share evoked components could help to define functional constraints on estimated solution. By leveraging the connections uncovered through cognitive mapping, we can construct robust source localization models with enhanced accuracy and interpretability.

Finally, we can address the issue of BCI illiteracy, which refers to the challenges faced by individuals to effectively operate BCIs [98, 99]. Our transfer learning approach, which could be applied at the subject level, holds promise in mitigating this challenge. BCI illiteracy is mostly observed on few specific tasks for a given user while accuracy on other tasks are correct [8]. Based on our results, a first possible approach is to apply task transfer, training on an effective decoding task for a user to transfer the results to an inefficient task. A second and more exploratory approach is to investigate user transfer, that is transfer the most effective decoding task from one subject and apply it to another with poor decoding abilities. This area presents an exciting and promising avenue for further exploration and improvement.

