# OpenReview forum: "Evaluating the structure of cognitive tasks with transfer learning"
_NeurIPS.cc/2023/Workshop/AI4Science — NeurIPS2023-AI4Science Poster_

### Official Review · Reviewer_ULo7 · 2023-10-24
**Comprehensive experiments but lack of novelty**

**Rating:** 5
**Confidence:** 4

**Review:**

The paper evaluated the transfer learning of two networks on two datasets. The comparisons included accuracy for the different tasks fine tuned from the proxy tasks. The paper clearly presented the formulas and the experiment settings. However, it lacks novelty. since it compared two already developed networks. The experiments were done on two datasets. Although they compared the accuracy on different sub - datasets, they only eveluate the transfering learning by accuracy, as shown in figures. It provided some explanation of the the structure of cognitive tasks by similary which were shown in the figures. I would recommend adding more datasets for comparison and make different settings on the proxy training task such as varying the amount of training datasets and training epochs.

---

### Meta-Review · Area_Chair_VHPh · 2023-10-27

**Recommendation:** Accept (Poster)
**Confidence:** 4

**Metareview:**

As noted by the reviewer, this work is experimental rather than methodological in nature as it relies on existing network architectures to examine cross paradigm transfer learning for EEG. The experimental validation is thorough, with a good amount of discussion on clinical/cognitive insights that such a study could provide for practical applications of EEG. Therefore, I am of the opinion that this work could result in follow up discussion that would be valuable to the relevant AI for Science community members.

COMMENTS/SUGGESTIONS Since the class prevalence statistics are not provided, it is unclear whether balanced accuracy is an appropriate metric for evaluating the efficacy of transfer learning. It would be good to comment on this choice in the appendix or provide a more detailed table of metrics.